# Oxidative Stress as an Important Contributor to the Pathogenesis of Psoriasis

**DOI:** 10.3390/ijms21176206

**Published:** 2020-08-27

**Authors:** Joanna Pleńkowska, Magdalena Gabig-Cimińska, Paweł Mozolewski

**Affiliations:** 1Department of Medical Biology and Genetics, University of Gdańsk, Wita Stwosza 59, 80-308 Gdańsk, Poland; joanna.plenkowska@phdstud.ug.edu.pl; 2Institute of Biochemistry and Biophysics, Polish Academy of Sciences, Laboratory of Molecular Biology, Kładki 24, 80-822 Gdańsk, Poland

**Keywords:** psoriasis, oxidative stress, reactive oxygen species/reactive nitrogen species, antioxidants

## Abstract

This review discusses how oxidative stress (OS), an imbalance between oxidants and antioxidants in favor of the oxidants, increased production of reactive oxygen species (ROS)/reactive nitrogen species (RNS), and decreased concentration/activity of antioxidants affect the pathogenesis or cause the enhancement of psoriasis (Ps). Here, we also consider how ROS/RNS-induced stress modulates the activity of transcriptional factors and regulates numerous protein kinase cascades that participate in the regulation of crosstalk between autophagy, apoptosis, and regeneration. Answers to these questions will likely uncover novel strategies for the treatment of Ps. Action in the field will avoid destructive effects of ROS/RNS-mediated OS resulting in cellular dysfunction and cell death. The combination of the fragmentary information on the role of OS can provide evidence to extend the full picture of Ps.

## 1. Introduction

About 3% of the world’s population suffers from psoriasis (Ps). Due to the fact that the number of patients is still growing but there are no effective therapies, it is important to know the exact molecular mechanism underlying this disease. Going through the latest reports on the pathogenesis of Ps, it is highlighted that one of the risk factors of this dermatosis is oxidative stress (OS) [1,2,3]. OS is caused by both increased production of reactive oxygen species (ROS)/reactive nitrogen species (RNS) and decreased concentration/activity of antioxidants that are responsible for their neutralization [4]. Although elevated ROS are signaling molecules to regulate biological and physiological processes, OS has been linked to a myriad of pathologies. OS, as a state of redox imbalance [5], leads to oxidative damage in cellular components (such as proteins, lipids, and nucleic acids), which may entail cellular disorders and induce cell death by apoptosis [6]. OS is one of the most important, quite often overlooked points in Ps. This complex, multifactorial syndrome is characterized by the occurrence of inflammatory infiltrates in hyperplasia and abnormally differentiated dermo-epidermal skin. In turn, human skin is a potential target for oxidative injury, as it is continuously exposed to environmental stimuli generating ROS [7]. Several conditions, such as infections, skin traumas, oxidant drugs, and stress factors, may cause and trigger the enhancement of Ps. It is generally assumed, that apart from psoriatic skin involvement, Ps is a systemic disease. In addition, it is frequently associated with some pathologies, namely cardiovascular dysfunctions, diabetes mellitus, and rheumatoid arthritis, which are known as “oxidative stress conditions” that may per se impose an oxidative stress condition. Furthermore, it has been suggested that there is an insufficient antioxidant system in the pathogenesis of Ps, resulting in a reduction of antioxidants, compounds that neutralize free radicals [8]. Although the importance of oxidant and antioxidant levels in the etiology or in the enhancement of Ps remains controversial, more and more findings, which are discussed in this review, support this hypothesis.

An oxidative stress condition typical for Ps is discussed in subsequent paragraphs and summarized in Figure 1. 

## 2. Oxidative Stress and Reactive Oxidative Species in Psoriasis

The pathogenesis of psoriasis (Ps) is still unclear, but recently the role of oxidative stress (OS) has been increasingly emphasized [9]. Moreover, oxidative stress constitutes an important factor affecting apoptosis [3] and, for instance, 17β-estradiol prevents apoptosis of human keratinocytes by reducing the level of oxidants [10]. Reactive oxidative species (ROS)- and reactive nitrogen species (RNS)-induced stress affects many physiological processes. Although high ROS/RNS concentrations primarily lead to cell death, low free radical levels can directly modulate the activity of transcriptional factors. ROS play a significant role in cell proliferation, differentiation, and death, including human keratinocytes and fibroblasts [11,12]. They also cause damage to many biomolecules in specific processes, including lipid peroxidation or stimulation of secretion proinflammatory cytokines [13]. ROS are produced under physiological conditions in the human body to play important roles in cell signaling and tissue homeostasis [14]. The abbreviation ROS refers to, among others, singlet oxygen, hydrogen peroxide, superoxide radicals, or hydroxyl radicals. They constitute a group of highly active molecules produced as by-products mainly from the respiratory activity of the OXPHOS system in mitochondria [15]. However, imbalance in the rate of ROS generation leads to oxidative stress and the production of free radicals that can damage DNA, proteins, and lipids [16]. Oxidized phospholipids can play an important role in many inflammatory diseases and mediate proinflammatory changes. Cells need low ROS level to function properly; however, excessive production is associated with disease states [17]. Moreover, the value of markers determining the level of oxidative stress and antioxidants are most often correlated with the severity of disease. [18]. This might be interesting in terms of finding further Ps markers. Notably, antioxidant delivery for the treatment of Ps has had positive effects [2]. The group of major antioxidants found in the skin includes catalase (CAT), superoxide dismutase (SOD), and glutathione peroxidase (GSH-Px). CAT, SOD, and GSH-Px belong to the group of enzymatic antioxidants. In addition to enzymatic antioxidants, non-enzymatic ones such as vitamin E, C, and glutathione (GSH) can also be distinguished. It should be emphasized that vitamin E (also called tocopherol) is involved in the creation of a physiological barrier, which makes it the most important non-enzymatic antioxidant [19]. Additionally, decreased α-tocopherol levels are observed in people with Ps [17]. As mentioned earlier, ROS participate in the protection of cells against pathogens or apoptosis with a properly functioning intracellular system. However, prolonged ROS action can lead to impaired antioxidant activity and damage of cellular components and skin tissue [20,21]. Whereas, the skin is constantly exposed to radiation and environmental factors that affect the release of reactive oxygen species. One of the most important functions of healthy skin is protection against harmful environmental factors. When damage occurs, inflammation develops, which, when properly regulated, initiates the healing process. Unfortunately, Ps is a chronic inflammation that is destructive to the human body [22]. An important point in the pathogenesis of Ps seems to be related to dysfunction in the antioxidant system, with increased production of ROS [23]. The exact mechanism of ROS action is still not fully understood and there is still no effective therapy against the harmful effects of ROS [21].

### 2.1. Effects of Oxidative Stress and Dyslipidemia on Skin Cells

The inflammation that occurs in Ps is associated with skin cells (keratinocytes and fibroblasts) and immune cells (both innate and adaptive) [24]. Interestingly, the intensity of ROS in the skin is definitely higher than in other tissues [25]. Oxidative stress also affects the formation of the damaged stratum corneum, which is crucial in psoriasis [3]. ROS are involved in the signaling pathways induced by TNFα [26]. While TNFα is an important factor in the pathogenesis of Ps, it leads to the formation of ROS in primary human keratinocytes, which in turn leads to the production of further cytokines [2]. Such cytokines as IL-4, IL-10, and EGF administered at low doses have reduced the level of oxidative stress in psoriatic fibroblasts. However, the exact mechanism of this reaction is still being sought [4]. It is possible that the relationship between cytokines and ROS may be important in the pathogenesis of skin diseases, including Ps [2]. Moreover, ROS increase the level of cytosolic Ca^2+^. Calcium and ROS interact with each other and a ROS concentration that is too high can lead to cell death through overloading due to high calcium level [25]. For proper functioning, the skin needs a calcium gradient, the appropriate level of which allows cell differentiation (high level of calcium) or proliferation (low level of calcium) [25]. ROS that are formed in keratinocytes and fibroblasts act chemotactically on neutrophils. Increased ROS production may also be associated with the accumulation of neutrophils in psoriatic lesions [27]. ROS derived from neutrophils, fibroblasts, and keratinocytes may be involved in the activation of neutrophils. In addition, ROS as a relay can be used to initiate NF-kB or AP-1 activation or production [9]. Interestingly, higher levels of ROS may appear in psoriatic fibroblasts, even before psoriatic plaques appear. The level of SOD is higher in fibroblasts, both from the lesional and nonlesional skin. In addition, research indicates the independence of lesions found in psoriatic fibroblasts from lymphocytes [28]. Moreover, NAPDH oxidase may be responsible for elevated levels of ROS in psoriatic fibroblasts [29]. A significant problem to investigate the exact mechanism of oxidative stress in psoriatic skin is the lack of a suitable animal model. Despite the many examples found in the literature, there is still a lack of animal models reflecting the actual condition of human psoriatic skin [2]. For instance, in literature we can find reports on a mouse psoriatic model under the control of ROS. Mannan’s team used this model as an activator of skin lesions typical of psoriasis [8]. Furthermore, ROS deficient mice had more skin lesions [8]. In Ps patients, damaged lipid metabolism in plasma is observed and there is a correlation between MDA and total cholesterol level [21]. The oxidation of polyunsaturated fatty acids caused by ROS are the cause of the formation of lipid peroxidation products (MDA) [21]. ROS may affect lipoprotein levels in people with Ps [22,30] and blood tests of patients indicate increased levels of lipid peroxidation markers [31,32]. There is a discussion on the relationship between Ps and cardiovascular/metabolic diseases due to changes in plasma lipoprotein levels. Attention is also paid to the relationship between the level of markers determining protein and lipid oxidation and the level of oxidative stress. Furthermore, low density lipoproteins were detected in lesional psoriatic skin [33]. The imbalance between the number of ROS and antioxidants causes lipid peroxidation, the formation of an oxidized fraction LDL (oxidized-LDL — ox-LDL) [34], and can lead to phospholipase A2 activation. However, the action of phospholipase is associated with the formation of arachidonic acid metabolites. During lipid peroxidation, cGMP is activated, while the cAMP level is reduced, resulting in excessive epidermal proliferation in Ps patients [27].

### 2.2. Antioxidant Enzymes (SOD, CAT, MDA, and GSH-Px) 

A crucial role in the process of antioxidant defense is played by enzymes, such as superoxide dismutase (SOD) and catalase (CAT), which are responsible for reducing the level of ROS [35,36]. SOD is necessary for proper function of the antioxidant system. It acts as a catalyst in the conversion of the superoxide anion into H_2_O_2_ and O_2_ [2]. SOD is considered the main defense mechanism, protecting against the toxic effects of oxygen [9], while CAT’s function is to decompose H_2_O_2_ into harmless substances, water, and oxygen [9]. There is a lack of clear information in the literature regarding the level of SOD in people with Ps. Some research teams report that the level of antioxidant enzymes is elevated in psoriatic fibroblasts, which decreases after treatment [37]. However, more often we can find information regarding the level of SOD in the epidermis of people with Ps. Conducted analyses indicate a decrease in the level of SOD in psoriatic keratinocytes. After using antioxidant supplementation, a restoration in the level of antioxidant enzymes can be observed corresponding to a healthy condition [38]. Similar results were obtained by the team of Young et al. They showed that the use of catalase or taurine in the treatment of Ps inhibits the expression of proinflammatory cytokines [26]. However, opinions about the effectiveness of antioxidant therapy are discussed [9]. Another study indicated the reduced level of SOD in the serum of people with Ps compared to the control group. Moreover, they explained that sometimes the increased value of SOD might be a result of compensatory activation [39]. Interestingly, information on SOD or CAT levels in Ps serum is variable and often we can read about reduced levels of both enzymes. However, there are also reports showing increased values of both enzymes [22,40,41]. Importantly, no correlations were observed with SOD levels in plasma and the severity of psoriasis [42]. The level of SOD and GSH-Px in psoriatic tissue is lower [43,44], while the values for CAT and MDA are significantly increased [19,41]. Moreover, the level of GPx (glutathione peroxidase) and CAT in erythrocytes of people with Ps was higher compared to the control group [45]. An increased level of malondialdehyde (MDA) and nitric oxide (NO) is observed, with a simultaneous decrease in the level of SOD and total antioxidant capacity (TAC) in the serum of patients with Ps [7,23]. Their levels correlate with the severity of Ps [23,46]. MDA testing allows for the assessment of fatty acid peroxidation [9] and MDA levels are increased in erythrocytes of people with Ps, while the levels of SOD and CAT in erythrocytes are reduced. Total antioxidant levels were lower in patients with active Ps compared to the group with an inactive disease at the time of the study [46]. Literature indicates that the level of oxidative stress correlates with the PASI index [19,36]. MDA, TOS, and OSI seems to be good biomarkers for Ps. However, research by Cannavò’s et al. showed that only MDA correlates with PASI [23]. It is likely that a more thorough analysis in the future will allow the use of antioxidants in Ps supportive therapy [18]. In January 2020, a publication appeared that also showed the importance of saliva in measuring oxidative stress levels in Ps patients. Measurements were made using saliva from affected people (stimulated and unstimulated) and from healthy individuals. Regardless of whether saliva was stimulated or not, an increase in GPx and CAT was observed compared to the control group. In addition, MDA (saliva, plasma) was increased in the studied group of patients [45]. The conducted analyses indicated a significant application in the case of TOS and OSI determination, both in stimulated and unstimulated saliva. Both parameters may, in the future, serve as diagnostic markers for determining the level of oxidative stress in Ps (Table 1). To our knowledge, for the first time the saliva from patients with Ps was used to assess the level of oxidative stress [45].

## 3. Perspective Markers of Oxidative Stress in Ps

### 3.1. Paraoxonase-1 

Paraoxoase-1 (PON1) belongs to the group of antioxidants (calcium dependent esterase) [56], is an enzyme associated with HDL, and has anti-inflammatory functions [56,57,58]. This enzyme has paraoxonase (PON) and arylesterase (ARE) activities [59]. The level of PON1 in patients with Ps was lower compared to the control group [60,61]; and, after implementation of Ps treatment, PON1 levels were higher compared to the control group. A high CRP in the patient’s plasma is associated with low PON1 levels [58]. Lowering PON1 levels in serum may be associated with significantly reduced LDL protection against oxidation [33,62,63]. Importantly, TNFα and IL-1 may also be responsible for reducing paraoxonase-1 level in serum [33,64]. Studies on a group of children have shown that Ps is associated with elevated MPO levels and decreased PON1 levels. The conducted analyses indicate HDL-related disorders. PON1 is located on the surface of this type of lipoprotein and, together with HDL, forms a protective structure against diseases associated with oxidative stress, especially against lipid peroxidation (both types HDL and LDL) [33,65]. PON1 activity does not correlate with PASI [66], but correlates with the level of apolipoprotein 1 [17,67]. However, sometimes PON1 activity may result from various polymorphic modifications. As reported by Asefi et al., patients with the PON1-L55M variant (conversion of leucine to methionine at position 55) may be at greater risk of psoriasis [17,68]. Furthermore, α-tocopherol supplementation may contribute to the normalization of PON1 [69]. To date, most PON1 studies have been conducted in Asia and no large-scale studies have been conducted in Europe [17]. Increased myeloperoxidase activity may affect PON1 and HDL levels [33]. Available information indicates increased MPO production by neutrophils in psoriatic lesions [33,70]. Furthermore, studies conducted on in vitro cultures confirm the relationship between MPO with ROS and lipid peroxidation (HDL and LDL) [33]. The elevated level of MPO was also present in the animal model of Ps [2,33].

### 3.2. Thiol/Disulphide

Thiols belong to 3 groups in plasma: protein, albumin, and low molecular weight (lower grade). In psoriasis, the thiol-disulfide balance is enhanced [71]. All of these contain a sulfhydryl group and counteract oxidative stress [27]. Cysteine and methionine are sulfur amino acids that are targets for various forms of oxygen. In human cells, the most common compound belonging to the thiol group is glutathione (GSH). The role of GSH is to create a suitable environment for redox reactions in cells [27,72]. Under the influence of ROS, thiols can transform into disulfides, which is a reversible reaction. This is crucial because of the ability to maintain disulfide-thiols homeostasis [73]. The thiol/disulfide level ratio is considered one of the biomarkers in oxidative stress [74]. It is believed that sulfur-containing compounds, including thiols, can protect against ROS [75].

### 3.3. MAP Kinases

MAP kinases are a group of serine-threonine kinases activated by mitogens [76]. All of them contain the Thr-Glu-Tyr (TXY) motif. Moreover, their activation requires phosphorylation of this motif [77]. Defects in the MAP kinases signaling pathway are assumed to be involved in Ps pathology. Experiments conducted by many research groups have confirmed that mitogen activated protein kinases, including p38, ERK1/2, and c-Jun kinase (JNK), are involved in the pathogenesis of Ps [78]. MAP kinases are important in the three signal pathways that control several functions in a cell, such as their proliferation, differentiation, gene expression, and apoptosis. ROS secretion may be the cause of MAP kinase activation in Ps, with particular emphasis on p38, Erk1/2, and JNK [79]. Erk1/2 kinase is associated with the response to growth factors, while JNK and p38 are associated with responses to cytokine levels and cellular stress. Furthermore, the literature indicates changes in the MAP kinase level when antioxidants or ROS scavengers are detected [80]. One of the Jun kinases (JunD) protects cells against the harmful effects of oxidative stress [81,82]. The ERK pathway in fibroblasts can be activated by reactive oxygen or nitrogen forms. The ERK1/2 pathway activated by oxidative stress can be inhibited with the PD98059 inhibitor. In some cases, the ERK1/2 kinase pathway may also be activated by the epidermal growth factor (EGF), secreted by the presence of ROS (RAS dependent pathway). There is also an independent RAS pathway for the activation of ERK1/2 kinase (e.g., through Src kinases). In addition, elevated calcium can also affect the level of ERK1/2 kinase. The literature indicates an elevated level of ERK1/2 kinase in people with Ps, which decreases during remission. This indicates that it may become a potential therapeutic target in this dermatosis [79]. The JNK pathway can be activated by various types of stressors, including UV radiation. It can also become activated, like ERK, by activating RAS. It is phosphorylated at tyrosine and threonine residues by MAPKK, which consequently leads to the activation of JNK kinase. Reactive oxygen species through the induction of ERK and ASK1, leads to the activation of JNK [79]. The p38 kinase is phosphorylated at a tyrosine and threonine residue [83]. Its path is activated by O_2_, H_2_O_2_, and NO [84]. The mechanism of this reaction is analogous to the JNK kinase, where ASK1 plays the main role [85]. Moreover, some studies indicate an increase in the level of p38 in psoriatic skin [79].

### 3.4. Tec Kinases

Dendritic cells and neutrophils are important in the formation of psoriatic lesions, especially in the early stages of the disease [24]. In addition, they are responsible for the formation of inflammatory infiltrates and microabscesses, characteristic of this type of skin dermatosis [86,87]. They also release oxidative mediators that increase inflammation in Ps [24]. Al-Hrabi et al. indicated an increase in the level of oxidative stress in neutrophils and CD11c + DC cells and concurrently, BT kinase (Burton’s tyrosine kinase) increased. They point to the possibility of using BTK as a therapeutic target in psoriasis because the use of a BTK inhibitor (Ibrutinib) allows a reduction in the level of oxidative stress [24]. Neutrophils secrete oxidants that have an effect on maintaining inflammation in psoriasis [88]. The BTK pathway in neutrophils increases the level of oxidative stress in the skin (iNOS induction) [89].

### 3.5. Sirtuin

Sirtuin 1 (SIRT1) belongs to a family of proteins that perform important functions in the process of metabolism or the cell cycle. Important from the point of view of Ps are their functions associated with inflammation, proliferation, or apoptosis [90]. SIRT1, as an NAD-dependent deacetylase, inhibits inflammation, excessive keratinocyte proliferation, and angiogenesis. In addition, its activation supports the inhibition of MAP or NF-kB kinase pathways, whose importance is in the process of oxidative stress and pathogenesis of Ps [90]. The literature indicates the significance of SIRT1 in the differentiation of keratinocytes, while inhibiting their hyperproliferation, as is the case with Ps. In addition, its use reduces the level of MAP kinases [90]. SIRT1 levels are lowered in psoriatic fibroblasts [91]. Studies conducted on a group of patients showed a reduction in SIRT1 level compared to the control group [92].

## 4. mTORC1 and Sestrins—Potential Stress Sensors in Ps

Recently, attention has been drawn to the mTORC1 (mammalian target of rapamycin complex 1) protein kinase as a regulator of epidermal homeostasis and its putative role in inflammatory skin diseases, including Ps [93,94,95,96]. The mTORC1 is a master growth regulator that senses diverse environmental cues, such as growth factors, cellular stresses, and energy levels. It has been shown that the mTORC1 is hyperactivated in lesional and nonlesional skin of psoriasis patients and its activity contributes to the proliferation of keratinocytes [94,95]. Moreover, mTORC1 activity has been identified as an important factor for several other types of skin cells [97]. Recently, sestrins (SESNs) have been identified as negative regulators of mTORC1 signaling through direct inhibition of GATOR2 complexes, which regulate trafficking to the lysosome membrane, thus altering kinase activity [98,99]. SESNs are a family of highly conserved proteins that are induced upon various cellular stresses, such as DNA damage, energy starvation, oxidative damage, and proinflammatory stimuli [100,101]. The crucial role of SESN proteins have been implicated in the UV responses of skin cells in terms of the suppression of tumorigenesis [102]. Moreover, it has been shown that SESNs are expressed in both keratinocytes and fibroblasts and they are upregulated during the UVB stress response [103]. Interestingly, after prolonged exposure to UV radiation, fibroblasts become more sensitive to oxidative stress [104]. Psoriatic fibroblasts may produce excessive amounts of H_2_O_2_, which leads to hyperproliferation of keratinocytes and NAPDH oxidase may be responsible for elevated levels of ROS in psoriatic fibroblasts [12]. Regarding the modulation of mTORC1 activity, it is also worth mentioning the role of metformin which suppresses immune responses by induction of AMPK and subsequent inhibition of mTORC1 [105,106,107]. Interestingly, metformin, which is the first-line medication in treating type 2 diabetes (T2DM), has been found to exert antiproliferative effect and has downregulated T helper-17 cells in experimental autoimmune animal models, as well as in autoimmune patients [108,109]. A very recent study has been shown that metformin treatment inhibited TNF-α- and IL-17A-induced inflammatory responses of keratinocytes by blocking NLRP3 inflammasome activation in keratinocytes [110]. Moreover, in another study, it has been shown that metformin suppresses the transcriptional activity of NF-κB in HaCaT cells and remarkably reduces inflammatory cytokines production, including TNFα, IL-6, IL-8, and IL-1β [111]. Taken together, these findings indicate that metformin-mediated anti-psoriatic effects on the skin could be a promising immunomodulatory add-on drug in Ps patients with T2DM.

## 5. Conclusions

ROS/RNS are produced in normal amounts as a part of the basic metabolism and play a role in numerous physiological mechanisms [19]. Disturbed proportions between the amount of oxidants and antioxidants lead to oxidative stress, while also causing lipid peroxidation. As a consequence, a series of consecutive reactions lead to cell damage [46]. The defense against free radicals are groups of enzymes (superoxide dismutase, catalase, peroxidases) and low molecular weight compounds (glutathione, ascorbic acid, β-tocopherol, and α-carotene) [80]. Importantly, in Ps, an increased level of total oxidative stress (TOS) is observed [23]. Thus, oxidative stress should be considered an important point in Ps [36]. Based on the available literature, research into oxidative stress in psoriasis is becoming more and more important. Oxidative stress in psoriasis leads to the activation of many signaling pathways (including NF-κB and MAPK) and, consequently, to the activation of Th1 and Th17 cells, the secretion of proinflammatory cytokines, then to hyperproliferation of keratinocytes, infiltration of immune cells into the skin, and changes in the permeability of blood vessels through lipid peroxidation. Currently, there is no complete and satisfactory cure for Ps, and numerous therapies can only alleviate some symptoms of the disease. The ultimate therapeutic goal is to restore the skin intact and relieve the systemic inflammation of this multifactorial disease. Therefore, many different options are currently investigated in order to improve therapeutic effects. Among them, methotrexate, cyclosporine, and oral retinoids are implemented, but the high dose and/or long-term usage of these agents still raises safety concerns [112]. On the other hand, due to its tolerable safety and high efficacy, biologics are now extensively investigated [113]. However, some cohort studies have shown that they are not effective to all individuals with Ps and may have some impact on comorbidities [114]. Thus, the choice of appropriate biologic therapy should be determined by the presence of coexisting conditions in patients with Ps. Alternatively, different non-pharmacological treatment approaches including lifestyle interventions such as body weight reduction or Mediterranean diet are considered to improve classical options [115,116]. Nonetheless, the limited data are now available on the impact of non-pharmacological interventions on Ps. One of the potential methods of treatment is antioxidants supplementation but the results of the research have yet to be seen. Perhaps treatment aimed at oxidative stress will become a chance for a better life for patients with psoriasis. It is possible that further research on oxidative stress in psoriasis will allow us to come one step closer to determining the mechanism of this disease.

## Figures and Tables

**Figure 1 ijms-21-06206-f001:**
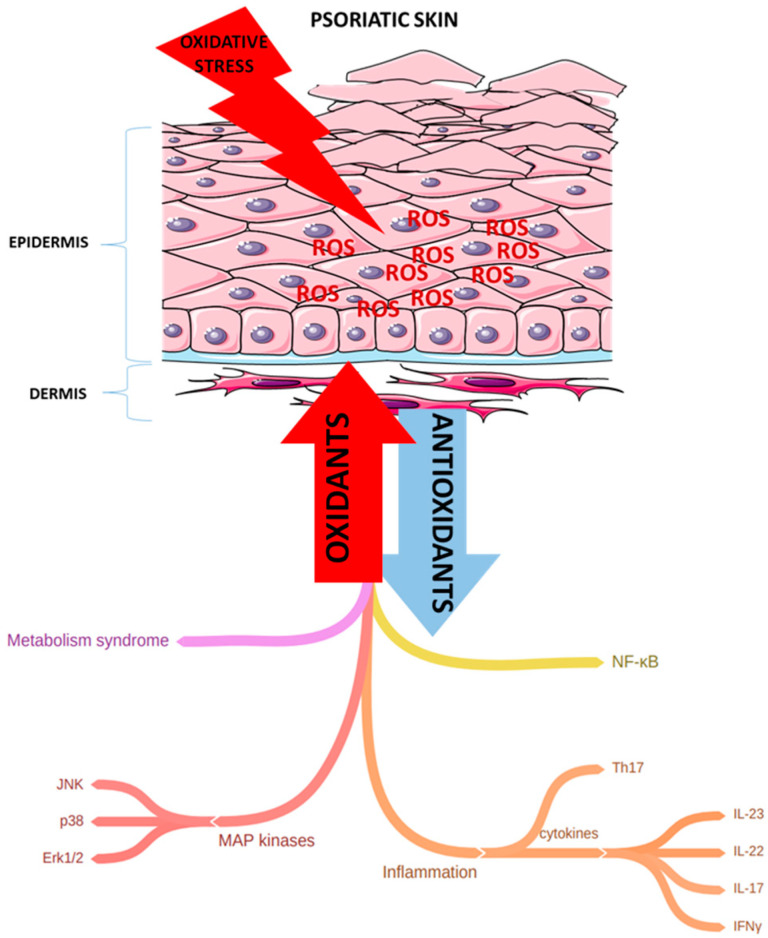
An oxidative stress condition typical for psoriasis. Oxidative stress leads to the overproduction of reactive oxidative species (ROS) that can damage DNA, proteins, lipids, and also results in the activation of many signaling pathways (including nuclear factor kappa-light-chain-enhancer—NF-κB and MAP kinase—mitogen-activated protein kinase), and in the stimulation of Th1 and Th17 cells and finally in the secretion of pro-inflammatory cytokines. All of this can lead to psoriatic inflammation. (JNK—c-Jun N-terminal kinase; ERK1/2—extracellular signal-regulated kinase 1/2; IFNγ—interferon γ; IL-23, IL-22, IL-17—interleukins: 23, 22, 17, respectively).

**Table 1 ijms-21-06206-t001:** The level of the main markers of oxidative stress in relation to the place of sample collection.

Lp.	Markers of Oxidative Stress		Fibroblasts	Keratinocytes	Serum	Plasma	Erythrocytes	Saliva	Author
**1**	ENZYMATIC	SOD	↑	↓	↑/↓	↓	↓	↑	Therond et al., 1996 [40]; Dimon-Gadal et al., 2000 [28]; Gornicki and Gutsze, 2001 [47]; Yildirim et al., 2003 [48]; Vanizor et al., 2003 [49]; Gerbaud et al., 2005 [37]; Kaharaeva et al., 2009 [38]; Pujari et al., 2010 [46]; Gabr and Al-Ghadir, 2012 [39]; Wagener et al., 2013 [22]
2		CAT	↑	↑	↑/↓	↓	↑/↓	↑	Thérond et al., 1996 [40]; Gornicki and Gutsze, 2001 [47]; Yildirim et al., 2003 [48]; Vanizor et al., 2003 [49]; Pujari et al., 2010 [46]; Skutnik-Radziszewska et al., 2020 [45]; Jarocka-Karpowicz et al., 2020 [50]
3		GSH-Px	↑	↓	n. d.	↑	↑/↓	↑	Thérond et al., 1996 [40]; Pujari et al., 2010 [46]; Kaur et al., 2016 [43]; Holmannova et al., 2020 [44]; Skutnik-Radziszewska et al., 2020 [45]; Jarocka-Karpowicz et al., 2020 [50]
4		MDA	↑	↑	↑	↑	↑	↑	Gornicki and Gutsze, 2001 [47]; Yildirim et al., 2003 [48]; Vanizor et al., 2003 [49]; Pujari et al., 2010 [46]; Gabr and Al-Ghadir, 2012 [39]; Şikar Aktürk et al., 2012 [51]; Skutnik-Radziszewska et al., 2020 [45]
5	NON-ENZYMATIC	Vitamin E	n.d.	n.d.	↓	↓	n.d.	n.d.	Pujari et al., 2010 [46]; Demir et al., 2013 [52]; Skutnik-Radziszewska et al., 2020 [45]; Oszukowska et al., 2020 [17]
6		GSH	↓	↓	↓	↓	n.d.	↓	Thérond et al., 1996 [40]; Asha et al., 2017 [53]; Taha and Al-Asady, 2019 [54]; Skutnik-Radziszewska et al., 2020 [45]; Jarocka-Karpowicz et al., 2020 [50]
7	ROS/RNS	H_2_O_2_	↑	↑	n.d.	n.d.	n.d.	n.d.	Dimon-Gadal et al., 2000 [28]; Hara-Chikuma and Satooka, 2016 [55]; Barygina et al., 2019 [12]
8		O_2_^•−^	↑	↑	n.d.	n.d.	n.d.	n.d.	Dimon-Gadal et al., 2000 [28]; Gabr and Al-Ghadir, 2012 [39]
9		NO^•^	↑	↑	↑	n.d.	n.d.	n.d.	Vanizor et al., 2003 [49]; Kadam et al., 2010 [36]; Barygina et al., 2019 [12]

↑ upregulated; ↓ downregulated; ↑/↓ divergent information; n.d.—no data; SOD—superoxide dismutase; CAT—catalase; GSH-Px—glutathione peroxidase; MDA—malondialdehyde; GSH—glutathione; H_2_O_2_—hydrogen peroxide; O_2_^•^^−^—superoxide radical; NO^•^—nitric oxide radical; ROS—reactive oxygen species; RNS—reactive nitrogen species.

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
