# Peer review of "Oxidative Stress as an Important Contributor to the Pathogenesis of Psoriasis"

_ijms, 2020, doi:10.3390/ijms21176206_

Round 1

Reviewer 1 Report

A very good manuscript, concisely and clearly describing the importance of oxidative stress in psoriasis. However, it is worth mentioning the importance of metformin in the treatment of psoriasis in the section "mTORC1 and Sestrins - potential stress sensors in Ps".

Author Response

Answer: We have added suitable information:

“Regarding the modulation of mTORC1 activity, it is also worth mentioning the role of metformin which suppresses immune responses by induction of AMPK and subsequent inhibition of mTORC1 [105-107]. Interestingly, metformin which is the first‐line medication in treating type 2 diabetes (T2DM), has been found to exert antiproliferative effect and has downregulated T helper-17 cells in experimental autoimmune animal models, as well as in autoimmune patients [108,109]. A very recent study has been shown that metformin treatment inhibited TNF-α- and IL-17A-induced inflammatory responses of keratinocytes by blocking NLRP3 inflammasome activation in keratinocytes [110]. Moreover, in another study, it has been shown that metformin suppresses the transcriptional activity of NF‐κB in HaCaT cells and remarkably reduces inflammatory cytokines production, including TNFα, IL‐6, IL‐8, and IL‐1β [111]. Taken together, these findings indicate that metformin-mediated anti-psoriatic effects on the skin could be a promising immunomodulatory add‐on drug in Ps patients with T2DM.”

Reviewer 2 Report

The authors present a review of the current state of knowledge of the role of oxidative stress in psoriasis.  The review is well-organized, clearly written, and appears to contain the most relevant and recent citations.  The only suggestion I have to improve the review is to include a brief discussion of the current therapies of psoriasis, and where the authors feel they fall short.  Other than that manuscript is acceptable for publication.  

Author Response

Answer: In the revised version of our manuscript, we have included additional text:

“Currently, there is no complete and satisfactory cure for Ps, and numerous therapies can only alleviate some symptoms of the disease. The ultimate therapeutic goal is to restore the skin intact and relieve the systemic inflammation of this multifactorial disease. Therefore, many different options are currently investigated in order to improve therapeutic effects. Among them, methotrexate, cyclosporine, and oral retinoids are implemented but the high dose and/or long‐term usage of these agents still raises safety concerns [112]. On the other hand, due to its tolerable safety and high efficacy, biologics are now extensively investigated [113]. However, some cohort studies have shown that they are not effective to all individuals with Ps and may have some impact on comorbidities [114]. Thus, the choice of appropriate biologic therapy should be determined by the presence of coexisting conditions in patients with Ps. Alternatively, different non-pharmacological treatment approaches including lifestyle interventions such as body weight reduction or Mediterranean diet are considered to improve classical options [115, 116]. Nonetheless, the limited data are now available on the impact of non-pharmacological interventions on Ps.”